# RECTIFYING GRADIENT-BASED OOD DETECTION VIA FISHER INFORMATION MATRIX

## ABSTRACT

Out-of-distribution (OOD) detection is an anomaly-handling mechanism, for which classification systems should detect outliers with true labels outside the label space, distinguishing them from normal in-distribution (ID) data. Advanced works suggest that gradient information preserves sufficient cues to indicate the confidence of being OOD. However, we discover previous gradient-based detection methods suffer from limited effectiveness mainly due to over-parameterization. As gradient-based OOD scores derive from the over-parameterized weight space, a widely recognized cause for the suboptimal OOD detection performance, there are also some gradient components which lack necessary information, thereby impairing the performance in OOD detection. This observation motivates us to propose *gradient rectification* (GradRect), using fisher information matrix to correct gradients in directions that are uninformative to discern the distribution change. Moreover, we connect GradRect with classical theories in identifying influential observations, verifying that model fine-tuning with outlier exposure can further improve GradRect. We conduct extensive experiments on various OOD detection setups, revealing the power of GradRect against state-of-the-art counterparts.

## 1 INTRODUCTION

Deep classification systems often encounter out-of-distribution (OOD) data whose true labels are not in the label space, and in such a situation classifiers cannot make right predictions as in-distribution (ID) data. This kind of phenomenon can lead to devastating results for many high-risk decision making applications (Li & Wechsler, 2005; Du et al., 2022; Shen et al., 2021). To address this issue, OOD detection aims to detect OOD cases to avoid making wrong predictions (Hendrycks & Gimpel, 2016; Liu et al., 2020; Sun et al., 2022; Djurisic et al., 2022), which can remarkably improve the reliability of deep learning in the open world (Wang et al., 2023).

OOD detection remains a challenging task, mainly owing to the calibration failures for modern deep models (Guo et al., 2017; Lee et al., 2018)—a well-trained ID classifier can make arbitrary-high softmax confidence on OOD data, making it unreliable in OOD detection. Accordingly, post-hoc OOD detection (Liu et al., 2020; Sun et al., 2022; Djurisic et al., 2022) aims at devising more accurate OOD indicators, *i.e.* scoring functions, other than softmax confidence. Among them, gradient-based scoring has received particular attentions (Huang et al., 2021; Igoe et al., 2022), which calculates gradient magnitudes *w.r.t.* model parameters to detect OOD data. Generally, model parameters should converge to local minimum for ID tasks with near-zero gradients after well training, while not for OOD data. Hence, the gradient magnitudes should preserve sufficient information to separate ID and OOD data, making gradient-based OOD detection a promising line of works.

Despite promising performance has been reported, these methods still exhibit certain limitations in practical applications. Inspired by the concept that deep models are susceptible to over-parameterization (Sun & Li, 2022; Djurisic et al., 2022), we clip GradNorm with varying percentages of gradient based on magnitudes and report the results of some OOD datasets on common CIFAR-100 and ImageNet benchmarks in Figure 1 (a) and (b). It is worth noting that on certain OOD datasets, as the percentage of clipped components increases, the performance of GradNorm has a slight improvement and then drop accordingly. This observation leads us to conjecture there exist some uninformative components in gradient which are redundant and detrimental for OOD

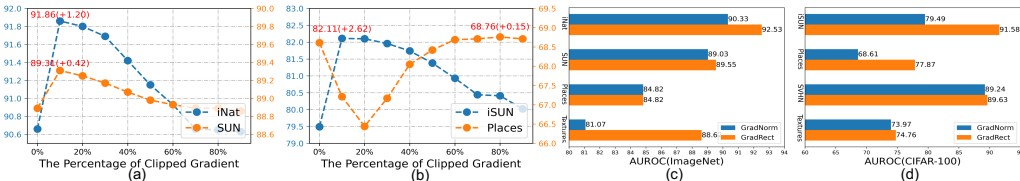

Figure 1: Plots showing (a) the performance of GradNorm with different percentage of clipped components on two OOD datasets (iNaturalist and SUN) of ImageNet-1k benchmark, (b) the performance of GradNorm with different percentage of clipped components on two OOD datasets (iSUN and Places365) of CIFAR-100 benchmark, (c) the performance of GradNorm and our method GradRect on ImageNet-1k benchmark, (d) the performance of GradNorm and our method GradRect on CIFAR-100 benchmark. (c) and (d) shows our GradRect achieves better and more stable performance than GradNorm.

detection. Even though simply clipping gradient based on magnitude indeed enhances performance on certain OOD datasets, this approach is impractical under common scenarios due to its case-sensitivity and the requirement for additional, subtle parameter-tuning operations. It may inadvertently discard crucial information for distinguishing between ID and OOD data.

To address this deficiency, this paper proposes *gradient rectification* (GradRect), a novel gradient-based scoring function which can rectify the original gradient information to more informative directions for OOD detection. GradRect rectifies original gradient based on the *fisher information matrix* (FIM) (Amari et al., 2019), which quantifies the amount of information carried for each direction in the gradient space. By multiplying gradient features with the inverse of the FIM, the adverse effects of uninformative component will be eliminated and the rectified gradients will align to directions that are more informative for OOD detection in the gradient space, leading to more effective and stable results on OOD detection than previous methods.

Theoretically, FIM equals to the average Hessian matrix on ID tasks under mild assumptions when using cross entropy loss as the objective for gradient calculation (Karakida et al., 2019). Then, the expression of GradRect is the magnitude of the influence function given the particular input (Koh & Liang, 2017), which estimates its effect on model predictions. Such a connection lead to two benefits. **(1)** We can interpret GradRect from the perspective in identifying influential observations: fitting OOD data has much larger influence on model parameters than ID ones, and thus influence function (and GradRect, equivalently) can effectively discern ID and OOD data. **(2)** We can justify that outlier exposure (Hendrycks et al., 2018), a particular fine-tuning approach, can further enhance GradRect: making ID (OOD) data with small (large) loss values will shrink (enlarge) the observing influence of ID (OOD) data, thus better separating GradRect and improving OOD detection.

We conduct experiments and establish superior performance on common OOD detection benchmarks, including classical CIFAR (Krizhevsky et al., 2009) and challenging ImageNet (Deng et al., 2009) benchmarks. Extensive evaluations show that our GradRect not only achieves superior performance over advanced OOD detection methods, but also exhibits greater stability across diverse tasks and backbones than previous gradient-based methods as shown in Figure 1 (c) and (d). Moreover, we perform ablation study among diverse backbone models and tasks, further demonstrating the stability of our method. We summarize our key contributions into four folds as follows:

- We offer new insights to make gradient-based scoring function more reliable in OOD detection and present the FIM as a useful tool to estimate the amount of information gradient carried in each direction *w.r.t.* the parameter space.

- We propose GradRect, which corrects original gradients to more stable and reliable directions through the FIM. Our proposed method compensates the drawbacks of instability compared to previous gradient-based detection methods in literature. We hope our method can draw further attention on exploiting gradients for OOD detection.

- Comprehensive experiments are carried out and show the superior performance of GradRect on various OOD detection benchmarks. Extensive experiments demonstrate the performance stability of GradRect on various model architectures and tasks.

- Theoretical interpretation is provided from the lens of classic influence function, explaining the basic mechanism behind GradRect. Inspired by this, we present that our GradRect can be combined with fine-tuning procedures, which further enhance the performance.

## 2 PRELIMINARIES

We begin by introducing necessary notations. Denote $\mathcal{X} \subseteq \mathbb{R}^d$ the input space and $\mathcal{Y} = \{1, 2, ..., C\}$ the label space, where $d$ is the input dimension and $C$ is the number of classes. Then, we discern the ID joint distribution $\mathcal{P}_{X_I Y_I}$ over $\mathcal{X} \times \mathcal{Y}$ and the OOD distribution $\mathcal{P}_{X_O}$ over $\mathcal{X}$. Therein, true labels of OOD data (i.e., $\boldsymbol{x}_O \sim \mathcal{P}_{X_O}$) are not in $\mathcal{Y}$, which are not predictable for the close-world models. Additionally, we possess a predictor $\boldsymbol{f_\theta} : \mathcal{X} \to \mathbb{R}^C$ (*i.e.*, logit outputs) parameterized by $\boldsymbol{\theta}$, typically trained on ID data that are i.i.d. drawn from $\mathcal{P}_{X_I Y_I}$ to make correct predictions.

**Out-of-distribution Detection.** During test, we may encounter a mixed distribution $\mathcal{P}_X$ of ID and OOD, defined as $\mathcal{P}_X = \alpha \mathcal{P}_{X_I} + (1 - \alpha) \mathcal{P}_{X_O}$ with $\alpha \in (0, 1)$ a mixing parameter. We typically employ the *scoring function* $S : \mathcal{X} \to \mathbb{R}$ to detect OOD data from ID ones: if $S(\boldsymbol{x})$ is larger than a threshold $\tau$, we will take the corresponding $\boldsymbol{x}$ as an ID case; otherwise an OOD case. Then, the question is how to find proper scoring functions for effective OOD detection.

**Scoring Functions.** We typically build the scoring functions upon our predictor $\boldsymbol{f_\theta}$. For example, as a well-known baseline scoring function, *maximum softmax prediction* (MSP) (Hendrycks & Gimpel, 2016) takes the confidence of label predictions in discerning ID and OOD cases, namely,

$$s_{\text{MSP}}(\boldsymbol{x}; \boldsymbol{\theta}) = \max_k \text{softmax}_k \boldsymbol{f_\theta}(\boldsymbol{x}), \tag{1}$$

where $\text{softmax}_k$ denotes the $k$-th elements of softmax outputs. Although straightforward, later works find that MSP often make mistakes in reality. Therefore, subsequent works focus on alleviating existing drawbacks for conventional MSP (Hendrycks & Gimpel, 2016) or proposing effective designing criteria for new scoring strategies (Liang et al., 2018; Lee et al., 2018; Liu et al., 2020).

## 3 GRADIENT-BASED OOD DETECTION

As a promising designing criterion, many works (Liang et al., 2017; Huang et al., 2021) use gradient information, calculated by backward propagation, to design new scoring functions. Gradients contain more information than outputs produced by forward propagation (Huang et al., 2021), potentially making gradient-based methods a promising line of work towards effective OOD detection.

### 3.1 PREVIOUS METHOD: GRADNORM

As a seminal work in gradient-based OOD detection, the GradNorm (Huang et al., 2021) leverages the gradient magnitudes in OOD scoring. It uses the Kullback-Leibler (KL) divergence between the softmax outputs and the uniform distribution $\boldsymbol{u} = [1/C, 1/C, \ldots, 1/C] \in \mathbb{R}^C$ as the objective, further calculating its gradients w.r.t. model parameters, namely,

$$s_{\text{GN}}(\boldsymbol{x}; \boldsymbol{\theta}) = \big\| \frac{\partial \ \text{KL}(\text{softmax}(\boldsymbol{f_\theta}(\boldsymbol{x})) \| \boldsymbol{u})}{\partial \ \boldsymbol{\theta}} \big\|_p, \tag{2}$$

where $\text{KL}$ is the KL divergence. Intuitively, gradient magnitudes should be larger than that for OOD data, for the reason where the models have trained to its minimum for ID cases, but not for OOD.

The GradNorm uses higher dimension features than conventional MSP as its inputs, thus containing more information in discerning ID and OOD features and leading to the improved detection performance. However, high dimension features also contain more components that are useless for OOD detection, causing the unstable and less-than-optimal results of GradNorm across different tasks. Therefore, we raise the following question: *Can we rectify the gradient features to discard those useless components to further improve gradient-based OOD detection?*

## 3.2 OUR METHOD: GRADRECT

To this end, we propose Gradient Rectification (GradRect), a novel gradient-based scoring strategy that further rectifies gradient features to improve post-hoc OOD detection. Our key mechanism relies on the Fisher information matrix (FIM) (Karakida et al., 2019), which is defined as follows.

**Definition 1 (Fisher Information Matrix (FIM))** *Considering the model $\boldsymbol{f_\theta}$ and the data distribution $\mathcal{P}_{XY}$, the fisher information matrix of $\boldsymbol{f_\theta}$ w.r.t. $\mathcal{P}_{XY}$ is defined by*

$$\mathcal{I}_{\mathcal{P}_{XY}}(\boldsymbol{\theta}) = \mathbb{E}_{\boldsymbol{x},\boldsymbol{y} \sim \mathcal{P}_{XY}} \frac{\partial l}{\partial \boldsymbol{\theta}} \frac{\partial l}{\partial \boldsymbol{\theta}}^\top , \tag{3}$$

*where $l := \log \mathrm{softmax}_{\boldsymbol{y}} \boldsymbol{f_\theta}(\boldsymbol{x})$ denotes the log-likelihood.*

As we can see, FIM is the covariance matrix for the derivative of the log-likelihood function, each element measures the amount of information within model parameters carried for the task defined by $\mathcal{P}_{XY}$ (Karakida et al., 2019). FIM is commonly used to assess the uncertainty associated with each parameter estimates, where a small magnitude of FIM in a direction indicates the associated components possesses higher sensitivity in the network, and thus the estimation is less precise.

**Using FIM.** When ID data are used to estimate FIM, i.e., $\mathcal{P}_{XY} = \mathcal{P}_{X_\mathrm{I} Y_\mathrm{I}}$, and the gradients of Kullback-Leibler (KL) divergence between the softmax outputs and a uniform distribution $\boldsymbol{u} = [1/C, 1/C, \ldots, 1/C] \in \mathbb{R}^C$ is adopted for OOD scoring, the directions in the gradient space with large magnitudes of FIM are more sensitive and less reliable in gradient-based OOD detection. Hence, it makes FIM an effective tool to indicate the misleading gradient information. To utilize FIM for gradient rectification, we suggest using FIM inverse for gradient rectification, following

$$\mathcal{I}^{-1}_{\mathcal{P}_{X_\mathrm{I} Y_\mathrm{I}}}(\boldsymbol{\theta}) * \frac{\partial \, \mathrm{KL}(\mathrm{softmax}(\boldsymbol{f_\theta}(\boldsymbol{x})) \| \boldsymbol{u})}{\partial \, \boldsymbol{\theta}}. \tag{4}$$

The intuition that motivates equation 4 is quite simple: since $\mathcal{I}_{\mathcal{P}_{X_\mathrm{I} Y_\mathrm{I}}}(\boldsymbol{\theta})$ is symmetric and positive definite, we have $\mathcal{I}_{\mathcal{P}_{X_\mathrm{I} Y_\mathrm{I}}}(\boldsymbol{\theta}) = U \Lambda U^{-1}$ via eigen decomposition (Brouwer & Eisenberg, 2018), with $U$ the orthogonal matrix that constructs the basis for the gradient space and $\Lambda$ a diagonal matrix whose diagonal elements indicate the basis variance. Accordingly, for each basis component $\boldsymbol{u}_i$ in $U$, a large $\lambda_i$ ($i$-th diagonal element in $\Lambda$) indicates the gradient direction following $\boldsymbol{u}_i$ is less informative and thus should be neglected. As $\mathcal{I}^{-1}_{\mathcal{P}_{X_\mathrm{I} Y_\mathrm{I}}}(\boldsymbol{\theta}) = U \Lambda^{-1} U^{-1}$, equation 4 first transforms the gradient vector $\partial \log \mathrm{softmax}\, \boldsymbol{f_\theta}(\boldsymbol{x})/\partial \boldsymbol{\theta}$ into the basis space by multiplying $U^{-1}$, then concentrating on those directions that are more informative by multiplying $\Lambda^{-1}$, finally recovering to the original gradient space by multiplying $U$. Therefore, using inverse FIM can correct gradient features, concentrating on gradient directions that are more informative for OOD detection.

**Estimating FIM.** FIM is defined by the correlation matrix for model gradients, where the expectation w.r.t. $\mathcal{P}_{XY}$ should be estimated in practice. Fortunately, based on the law of large numbers (Judd, 1985), one can simply derive a consistent and unbiased estimator for $\mathcal{I}_{\mathcal{P}_{X_\mathrm{I} Y_\mathrm{I}}}(\boldsymbol{\theta})$ when having $N$ *i.i.d.* input-output pairs $(x_1, y_1), \ldots, (x_N, y_N)$, leading to the empirical FIM following

$$\hat{\mathcal{I}}_{\mathcal{P}_{X_\mathrm{I} Y_\mathrm{I}}}(\boldsymbol{\theta}) = \frac{1}{N} \sum_{i=1}^N \frac{\partial l_i}{\partial \boldsymbol{\theta}} \frac{\partial l_i}{\partial \boldsymbol{\theta}^\top}. \tag{5}$$

To compute $\hat{\mathcal{I}}_{\mathcal{P}_{X_\mathrm{I} Y_\mathrm{I}}}(\boldsymbol{\theta})$ in practice, we mitigate biases and outliers by selecting training data with their confidence scores above a certain threshold, considering these data are well trained and representative. Moreover, similar to previous works (Huang et al., 2021), we do not compute the gradients w.r.t. all model parameters, instead considering only the last fully connected layer to save computation costs. Note that $\hat{\mathcal{I}}_{\mathcal{P}_{X_\mathrm{I} Y_\mathrm{I}}}(\boldsymbol{\theta})$ and its inverse can be calculated in advance, thus without introducing much additional computational costs when calculating GradRect.

**GradRect Scoring.** We now define our gradient-based scoring function that harnesses gradient rectification as in equation 4. Therein, we estimate FIM via equation 5 and gradient magnitude for OOD scoring following equation 2. To sum up, our GradRect scoring function is given by

$$s_{\mathrm{GradRect}}(\boldsymbol{x}; \boldsymbol{\theta}) = \| \hat{\mathcal{I}}^{-1}_{\mathcal{P}_{X_\mathrm{I} Y_\mathrm{I}}}(\boldsymbol{\theta}) * \frac{\partial \, \mathrm{KL}(\mathrm{softmax}(\boldsymbol{f_\theta}(\boldsymbol{x})) \| \boldsymbol{u})}{\partial \, \boldsymbol{\theta}} \|_p, \tag{6}$$

where we use the $L_p$ norm for rectified gradients as the scoring function. Following (Huang et al., 2021), we assume $p = 2$ by default. For other choices of $p$, please refer to Appendix. By rectifying original gradient based on FIM, we improves the separation of ID and OOD data, as shown in Figure 3, thereby enhancing the performance of OOD detection. The overall framework of GradRect is summarized in Figure 2.

### 3.3 Outlier Exposure Can Improve GradRect

We have demonstrated that GradRect can be used for post-hoc OOD detection given a well-trained ID classification model, while further improvement with model fine-tuning is also of our interest. Specifically, we study if we can suggest a specific model training scheme, of which the resultant model can further improve GradRect in OOD detection.

**Outlier Exposure.** We find that conventional outlier exposure (OE) (Hendrycks et al., 2019), which originally aims at maximizing the MSP score, can already be used to improve GradRect. Overall, OE makes the model learn to discern ID and OOD data by using the OOD distribution $\mathcal{P}_{X_O}$ during training. Specifically, OE adopts the learning objective as follows:

$$\mathbb{E}_{(\boldsymbol{x},y)\sim\mathcal{P}_{X_I Y_I}} - \log \text{softmax}_y \, \boldsymbol{f_\theta}(\boldsymbol{x}) + \lambda \mathbb{E}_{\boldsymbol{x}\sim\mathcal{P}_{X_O}} - \text{KL}(\text{softmax}(\boldsymbol{f_\theta}(\boldsymbol{x}))\|\boldsymbol{u}), \quad (7)$$

where $\lambda$ is the trade-off parameter. The first term in equation 7 makes the model produce high softmax confidence for ID data while the second term makes the model produce low softmax confidence for OOD data. Although simple, OE remains one of the most effective fine-tuning scheme to improve OOD detection, while the studies for gradient-based OOD scoring are limited.

**Influence Function and GradRect.** The key to demonstrate OE can improve GradRect is the link between influence function and GradRect. In the context of machine learning, the influence function is used to estimate the effect of individual training examples on a model's predictions. It quantifies the influence of each training example on the model's output, which is defined as the change of parameters $\boldsymbol{\theta}$ when a training data $\boldsymbol{z} = (\boldsymbol{x}, y)$ is upweighted by some small $\epsilon$, following:

$$\boldsymbol{\theta_{z,\epsilon}} = \arg \min_{\boldsymbol{\theta}} \mathbb{E}_{(\boldsymbol{x}_i,y_i)\sim\mathcal{P}_{X_I Y_I}} - \log \text{softmax}_{y_i} \, \boldsymbol{f_\theta}(\boldsymbol{x}_i)$$
$$+\epsilon(- \log \text{softmax}_y \, \boldsymbol{f_\theta}(\boldsymbol{x})). \quad (8)$$

The classical statistical theorems (Ling, 1984) have told us that the influence of upweighting $\boldsymbol{z}$ on the parameter $\boldsymbol{\theta}$ is given by:

$$\frac{\partial \boldsymbol{\theta_{z,\epsilon}}}{\partial \epsilon} \Big|_{\epsilon=0} = -\mathcal{H}_{\mathcal{P}_{X_I Y_I}}^{-1}(\boldsymbol{\theta}) * \frac{\partial \log \text{softmax}_y \, \boldsymbol{f_\theta}(\boldsymbol{x})}{\partial \boldsymbol{\theta}}, \quad (9)$$

where $\mathcal{H}_{\mathcal{P}_{X_I Y_I}}(\boldsymbol{\theta})$ is the Hessian matrix. Moreover, under certain regularity conditions (Lehmann & Casella, 2006), we further have $\mathcal{H}_{\mathcal{P}_{X_I Y_I}}(\boldsymbol{\theta}) = \mathcal{I}_{\mathcal{P}_{X_I Y_I}}(\boldsymbol{\theta})$, which is exactly the rectified gradient defined in equation 4. Thus, our GradRect can also be interpreted from the perspective in identifying those influential observations *w.r.t.* the KL loss and ID data. Therefore, our GradRect can effectively distinguish between ID and OOD cases.

**Outlier Exposure and Influence Function.** Note that equation 7 exactly maximizes the influence function defined by equation 8. When equation 7 is minimized, the first term will encourage ID data to have less influence on model parameters, since the model after OE can work well on the ID classification task. By contrast, the second term will encourage OOD data to have more influence *w.r.t.* the ID classification task, since OOD data with any label in $\mathcal{Y}$ are not fitted by the model and will change the model parameters a lot. Therefore, after OE training, the model can distinguish between the influence of ID and OOD cases, improving GradRect in OOD detection due to equation 9.

## 4 Experiments

We describe the experiment details in Section 4.1 including baseline models, pre-training setups and evaluation metrics. Then, in Section 4.2, we report the superior performance of our GradRect against state-of-the-arts on both the CIFAR (Krizhevsky et al., 2009) and the ImageNet (Deng et al., 2009) benchmarks. In Section 4.3, we further conduct extensive ablation studies and more analysis.

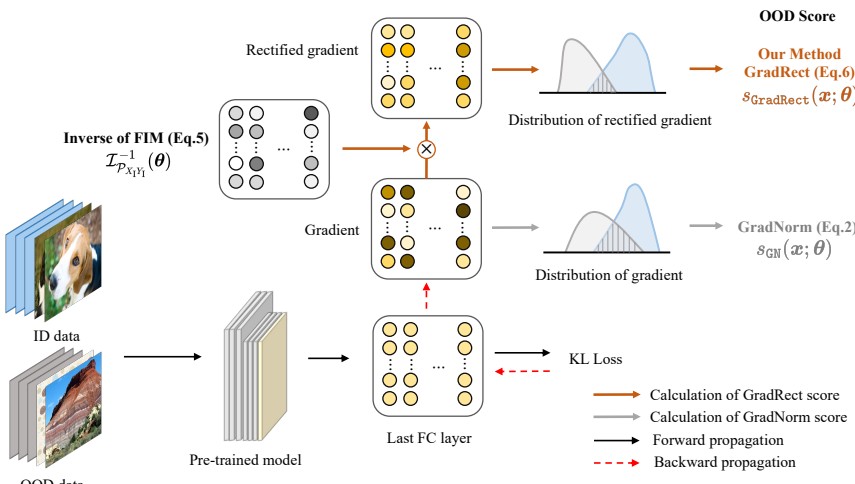

Figure 2: Illustration of our framework using GradRect for OOD detection. Our GradRect rectifies the original gradient from the last fully connected (FC) layer based on FIM. After rectification, the adverse effect of uninformative components in gradient is eliminated, resulting in stronger separability between ID and OOD data.

## 4.1 IMPLEMENTATION DETAILS

**Baseline Methods.** We compare our GradRect with advanced methods in OOD detection. We mainly consider baseline methods which can be directly applied to the pre-trained model. We compare GradRect with seven recent post-hoc OOD detection methods, namely MSP (Hendrycks & Gimpel, 2016), ODIN (Liang et al., 2018), Mahalanobis (Lee et al., 2018), Energy (Liu et al., 2020), DICE (Sun & Li, 2022), ReAct (Sun et al., 2021), and GradNorm (Huang et al., 2021). As stated in 3.3, our method can combine with other fine-tuning methods to further improve the performance. For fair comparison, we select six advanced methods which need retraining the model to compare, including SSD+ (Sehwag et al., 2021), methods that use an auxiliary outlier dataset but randomly select outliers during training, namely OE (Hendrycks et al., 2018), CSI (Tack et al., 2020), SOFL (Mohseni et al., 2020), CCU (Meinke & Hein, 2019), and methods involving outlier mining, namely NTOM (Chen et al., 2021) and POEM (Ming et al., 2022).

**Pre-training Setups.** We conduct experiments on the large-scale ImageNet (Deng et al., 2009) and CIFAR (Krizhevsky et al., 2009) benchmarks. The large-scale ImageNet dataset is more challenging than the traditional CIFAR benchmark, primarily due to the notably larger and more diverse image space. For the ImageNet case, we select four testsets from subsets of iNaturalist (Van Horn et al., 2018), SUN (Xu et al., 2015), Places365 (Zhou et al., 2017), and Texture (Cimpoi et al., 2014), which are craft by (Huang & Li, 2021) with non-overlappping categories *w.r.t.* ImageNet. For the CIFAR cases, we employ Texture (Cimpoi et al., 2014), SVHN (Netzer et al., 2011), Places365 (Zhou et al., 2017), LSUN-Crop (Yu et al., 2015), LSUN-Resize (Yu et al., 2015), and iSUN (Xu et al., 2015).

In terms of models, we use the Google BiT-S (Kolesnikov et al., 2020) pretrained on ImageNet-1k with ResNetv2-101 architecture (He et al., 2016) on the ImageNet case, and use a pretrained DenseNet-101 architecture following the setting in Sun & Li (2022) on the CIFAR cases.

**Evaluation Metrics.** The OOD detection performance is evaluated via two common metrics, which are both threshold-independent (Davis & Goadrich, 2006): the false positive rate of OOD data when the true positive rate of ID data is at $95\%$ (FPR95); and the *area under the receiver operating characteristic curve* (AUROC), which is the probability of ID case having greater score than OOD, depicting the relationship between true positive rate and false positive rate.

## 4.2 RESULTS

We present the main results on ImageNet and CIFAR benchmarks.

Table 1: Comparison in OOD detection on the ImageNet benchmark. Baseline methods include post-hoc methods. ↓ (or ↑) indicates smaller (or larger) values are preferred. Bold font indicates the best results in a column.

| Method | iNaturalist | | SUN | | Places | | Texture | | Average | |
|---|---|---|---|---|---|---|---|---|---|---|
| | FPR95 ↓ | AUROC ↑ | FPR95 ↓ | AUROC ↑ | FPR95 ↓ | AUROC ↑ | FPR95 ↓ | AUROC ↑ | FPR95 ↓ | AUROC ↑ |
| MSP (Hendrycks & Gimpel, 2016) | 63.69 | 87.59 | 79.89 | 78.34 | 81.44 | 76.76 | 82.73 | 74.45 | 76.96 | 79.29 |
| ODIN (Liang et al., 2018) | 62.69 | 89.36 | 71.67 | 83.92 | 76.27 | 80.67 | 81.31 | 76.30 | 72.99 | 82.56 |
| Mahalanobis (Lee et al., 2018) | 96.34 | 46.33 | 88.43 | 65.20 | 89.75 | 64.46 | 52.23 | 72.10 | 81.69 | 62.02 |
| Energy (Liu et al., 2020) | 64.91 | 88.48 | 65.33 | 85.32 | 73.02 | 81.37 | 80.87 | 75.79 | 71.03 | 82.74 |
| ReAct (Sun et al., 2021) | 49.97 | 89.80 | 65.30 | 87.40 | 73.12 | **85.34** | 80.82 | 70.53 | 67.30 | 83.27 |
| GradNorm (Huang et al., 2021) | 50.03 | 90.33 | 46.48 | 89.03 | 60.86 | 84.82 | 61.42 | 81.07 | 54.70 | 86.71 |
| GradRect | **38.56** | **92.53** | **46.35** | **89.55** | **58.44** | 84.82 | **44.96** | **88.62** | **47.08** | **88.88** |

### 4.2.1 IMAGENET BENCHMARK

The large-scale ImageNet benchmark can provide clues about model performance in real-world applications due to its large semantic space with about 1k classes. In Table 1, we compare the results of GradRect with advanced methods on ImageNet and report performance for each OOD test dataset, as well as the average performance. GradRect achieves state-of-the-art performance on all metrics, with 47.08% FPR95 and 88.88% AUROC. Compared to GradNorm (Huang et al., 2021), GradRect improves the FPR95 by 7.62% and AUROC by 2.17%, demonstrating the effectiveness of our method in large-scale applications.

Moreover, as visualized in Figure 3, by mitigating the influence of uninformative components in gradient, our proposed GradRect score produces better distinguished distributions, validating the effect of gradient rectification in OOD detection. It is worth noting that performance differences exist among various OOD datasets. We believe that the reason behind this disparity lies in the fact that for OOD datasets with distributions that are further away from the ID data, the gradient information is less influenced by the quality of the model's parameters .

It's notable that there is the performance difference between various OOD datasets and we consider the reason is that for those OOD datasets which have distribution more similar to ID data, their gradient information is more affected by the quality of model's parameters. Consequently, the rectification can enhance performance more significantly.

### 4.2.2 CIFAR BENCHMARKS

**Comparison with post-hoc methods.** For both the CIFAR-10 and CIFAR-100 benchmarks, we report the detection performance for the average of the six OOD datasets. As shown in Table 2, our method GradRect can lead to more effective OOD detection performance than GradNorm and outperforms most of the baselines considered. For example, on the CIFAR-100 benchmark, GradRect improves the AUROC by 1.12% when compared to GradNorm and reduces the FPR95 by 7.52% when compared to the previous best method DICE (Sun & Li, 2022), demonstrating the superiority of our method on different OOD situations.

**Comparison with fine-tuning methods.** Considering the connection between influence function and GradRect, methods that leverages auxiliary outlier data can further improve the performance of GradRect as stated in Section 3.3. Therefore, we combine GradRect with the fine-tuning method in POEM (Ming et al., 2022) and report results in Table 3. It is shown that GradRect can be significantly improved via the combination with fine-tuning and outperforms previous fine-tuning approaches by remarkable margins.

### 4.3 ABLATION STUDY

We now conduct ablation studies from various aspects to further improve our understandings.

**Effect of alternative neural network architectures.** To further investigate the effectiveness and robustness of our method, we perform OOD detection on remaining two architectures, ResNet-50 and MobileNetV2, both of which are trained with ID data (ImageNet-1k) only. The results over four datasets and the average of four are shown in Table 4. The accuracy on ID datasets, number of parameters and improvement of GradRect compared to GradNorm (AUROC ↑) for ResNetv2-101 is 75.19%, 44.5M and +2.17; for ResNet-50, it's 76.13%, 26M and +0.99; for MobileNetV2,

Table 2: Comparison in OOD detection on the CIFAR benchmarks with post-hoc methods. Bold font indicates the best results in a column.

| Method | CIFAR-10 | | CIFAR-100 | |
|---|---|---|---|---|
| | FPR95 ↓ | AUROC ↑ | FPR95 ↓ | AUROC ↑ |
| Post-hoc Approaches | | | | |
| MSP (Hendrycks & Gimpel, 2016) | 48.73 | 92.46 | 80.13 | 74.36 |
| ODIN (Liang et al., 2018) | 24.57 | 93.71 | 58.14 | 84.49 |
| Mahalanobis (Lee et al., 2018) | 31.42 | 89.15 | 55.37 | 82.73 |
| Energy (Liu et al., 2020) | 26.55 | 94.57 | 68.45 | 81.19 |
| ReAct (Sun et al., 2021) | 26.45 | 94.95 | 62.27 | 84.47 |
| DICE (Sun & Li, 2022) | 20.83 | 95.24 | 49.72 | 87.23 |
| GradNorm (Huang et al., 2021) | 21.30 | 95.08 | 49.73 | 86.48 |
| GradRect | **19.78** | **95.41** | **42.20** | **87.60** |

Table 3: Comparison in OOD detection on the CIFAR benchmarks with fine-tuning methods. † denotes the method with fine-tuning. Bold font indicates the best results in a column.

| Method | CIFAR-10 | | CIFAR-100 | |
|---|---|---|---|---|
| | FPR95 ↓ | AUROC ↑ | FPR95 ↓ | AUROC ↑ |
| Fine-tuning Approaches | | | | |
| SSD+ (Sehwag et al., 2021) | 7.22 | 98.48 | 38.32 | 88.91 |
| OE (Hendrycks et al., 2018) | 9.66 | 98.34 | 19.54 | 94.93 |
| SOFL (Mohseni et al., 2020) | 5.41 | 98.98 | 19.32 | 96.32 |
| CCU (Meinke & Hein, 2019) | 8.78 | 98.41 | 19.27 | 95.02 |
| NTOM (Chen et al., 2021) | 4.38 | 99.08 | 19.96 | 96.29 |
| POEM (Ming et al., 2022) | 2.54 | 99.40 | 15.14 | 97.79 |
| GradRect † | **2.37** | **99.51** | **4.22** | **99.04** |

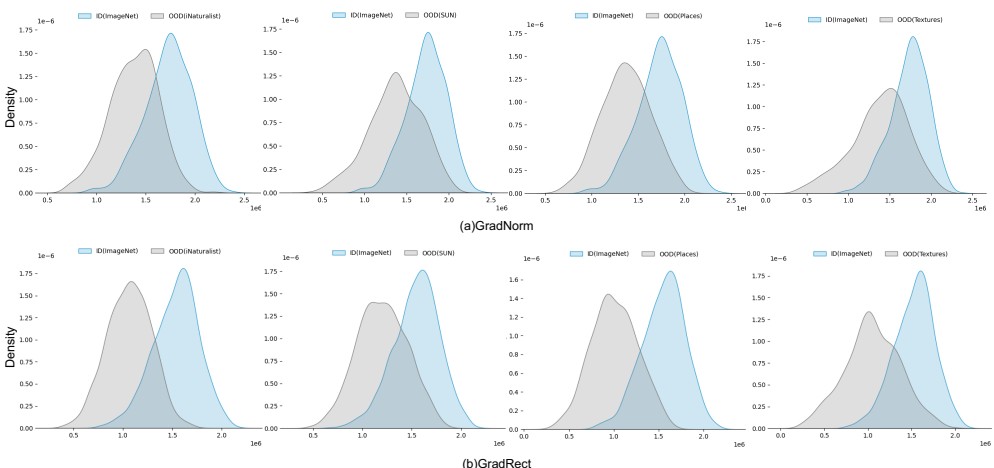

Figure 3: Plots showing the distribution of GradNorm and GradRect scores on ID (ImageNet) dataset and four OOD datasets, which illustrates our GradRect applying rectification to gradient by FIM can enhance the separation between ID and OOD data.

is 71.88%, 3.5M and +0.26. It is apparent that our score can gain comparable improvement on various architectures. Additionally, the performance on different backbones is mainly influenced by the model's parameter quantity and accuracy. Before the model reaches the overfitting stage, the gradient remains fluctuating and our method excels at mitigating the instability direction in gradient, improving performance more effectively. However, as it approaches overfitting, gradient becomes more uninformative and disordered, which is hard to rectify by a certain measurement. This explains the performance difference between various architectures.

**Effect of gradient of different parameters.** In this ablation, we investigate several variants of GradRect where the gradients are from different network layers. In line with GradNorm, we consider gradients from four strategies (1) all parameters: all trainable parameters from all layers of the network, (2) parameters from block n: all trainable parameters in the $n$-th block, (3) parameters from the last layer: parameters from the last fully connected (FC) layer.

From the results in Table 5, we can get the similar conclusion as in GradNorm that the gradient information from deeper layers can perform better on OOD detection than shallower layers, mainly due to the reason that the gradients from deeper layers preserve more information of data thus can be more effectively distinguish OOD data from ID data. It is noted that gradient from the last fully connected layer yield the best performance and is computationally convenient in practice. Therefore, we only leverage GradRect with gradient from the last FC layer in the experiments.

**Effect of the proportion of rectified gradient.** In this ablation, we investigate the impact of varying the proportion of rectified gradient and present the corresponding results in Table 6. Remarkably, as the proportion of the rectified component increases, a notable improvement in performance is ob-

Table 4: Comparison of GradRect and advanced methods with different architectures on the ImageNet benchmark. ↓ (or ↑) indicates smaller (or larger) values are preferred. Bold font indicates the best results in a column.

| Structure | Param. | ID Acc. | Method | iNaturalist | | SUN | | Places | | Texture | | Average | |
|---|---|---|---|---|---|---|---|---|---|---|---|---|---|
| | | | | FPR95↓ | AUROC↑ | FPR95↓ | AUROC↑ | FPR95↓ | AUROC↑ | FPR95↓ | AUROC↑ | FPR95↓ | AUROC↑ |
| **ResNet-50** | 26M | 76.13 | Energy Liu et al. (2020) | 55.72 | 89.95 | 59.26 | 85.89 | 64.92 | 82.86 | 53.72 | 85.99 | 58.41 | 86.17 |
| | | | GradNorm Huang et al. (2021) | 44.10 | 89.37 | 44.10 | 90.38 | 56.50 | 85.67 | 51.10 | 88.78 | 48.95 | 88.55 |
| | | | GradRect | **39.50** | **91.08** | **39.90** | **91.34** | **54.70** | **86.92** | 57.30 | 88.82 | **47.85** | **89.54** |
| **MobileNetV2** | 3.5M | 71.88 | Energy Liu et al. (2020) | 59.50 | 88.91 | 62.65 | 84.50 | 69.37 | 81.19 | 58.05 | 85.03 | 62.39 | 84.91 |
| | | | GradNorm Huang et al. (2021) | 35.90 | 91.81 | 45.50 | 90.30 | 57.70 | 86.13 | 38.90 | 91.98 | 44.50 | 90.05 |
| | | | GradRect | **34.80** | **91.94** | **44.30** | **90.95** | **54.70** | **86.24** | **38.30** | **92.22** | **43.03** | **90.31** |

Table 5: Effect of gradient from different part of parameters (ResNetv2-101 on ImageNet-1k). The results show gradient norm derived from deeper layers yield better performance.

| Gradient Space | FPR95↓ | AUROC↑ |
|---|---|---|
| Block1 | 72.95 | 77.02 |
| Block2 | 70.46 | 79.51 |
| Block3 | 67.93 | 80.85 |
| Block4 | 61.86 | 86.46 |
| All Params | 67.89 | 82.37 |
| Last Layer Params | 47.08 | 88.88 |

Table 6: Effect of the proportion of rectified gradient on ImageNet-1k benchmark. ∆ denotes the improvement. The result highlights the effectiveness of gradient rectification.

| Proportion | FPR95↓ | AUROC↑ | ∆ FPR95↓ | ∆ AUROC↑ |
|---|---|---|---|---|
| No Rect. | 54.70 | 86.71 | - | - |
| 0.01 | 51.92 | 87.56 | -2.78 | +0.85 |
| 0.05 | 49.67 | 88.64 | -5.03 | +1.93 |
| 0.1 | 47.08 | 88.88 | -7.62 | +2.17 |
| 0.3 | 46.73 | 89.10 | -7.97 | +2.39 |
| 0.5 | 45.95 | 89.10 | -8.75 | +2.39 |
| 0.9 | 46.20 | 89.11 | -8.5 | +2.40 |
| all | 45.90 | 89.08 | -8.8 | +2.37 |

served. This progressive enhancement in performance serves as compelling evidence to substantiate the effectiveness of gradient rectification in effectively discriminating between ID and OOD data.

**Effect of rectified gradient in optimization.** Our method also has relationship with natural gradient descent in optimization which can be seen as a type of 2nd-order optimization method. Through multiplying gradient with FIM, natural gradient descent works by performing a local quadratic approximation to the objective around the current iterate and can make much more progress given a limited iteration budget compared to traditional stochastic gradient descent Martens (2020). A toy example of loss function is shown in Fig 4, from which we can observe gradient rectified by FIM gains rapid exploration of low-curvature directions and thus faster convergence in optimization.

## 5 RELATED WORK

In this section, we briefly review the related works in OOD detection and influence function.

**OOD Detection.** In real-world deployment, neural network need to figure OOD data which have non-overlapping label space with training data instead of giving wrong predictions, which gives rise to the importance of OOD detection. The phenomenon of the overconfidence in OOD data in neural network is first revealed in Nguyen et al. (2015). To solve this problem, a plethora of algorithms designed for the detection of OOD data have been proposed.

Based on whether the model needs to be fine-tuned, existing algorithms can be roughly categorized into two types, post-hoc methods and fine-tuning methods. The post-hoc methods aim to detect OOD data without additional training procedures and could be directly applied to any pre-trained models. A main stream of work in this category is to devise a scoring function based on the intermediate or the final output of the model, such as OpenMax score Bendale & Boult (2015), Maximum Softmax Probability Hendrycks & Gimpel (2016), ODIN score Liang et al. (2017), Energy score Liu et al. (2020); Lin et al. (2021); Wang et al. (2021), Activation rectification (ReAct) Sun et al. (2021), and ViM score Wang et al. (2022). Another line of work focus on exploiting feature space to discern OOD data, including Mahalanobis distance-based score Lee et al. (2018), non-parametric KNN-based score Sun et al. (2022). The concept of utilizing gradient information to assist OOD detection is first introduced by ODIN score Liang et al. (2017). Their method involves a pre-processing which adds perturbations derived from gradient to input data, increasing the distinction between the softmax scores of ID data and OOD data and resulting in better performance. More recently, Huang *et al.* Huang et al. (2021) proposed to design scoring function based on gradient norms, which is motivated by the observation that ID and OOD data tend to exhibit different gradient patterns

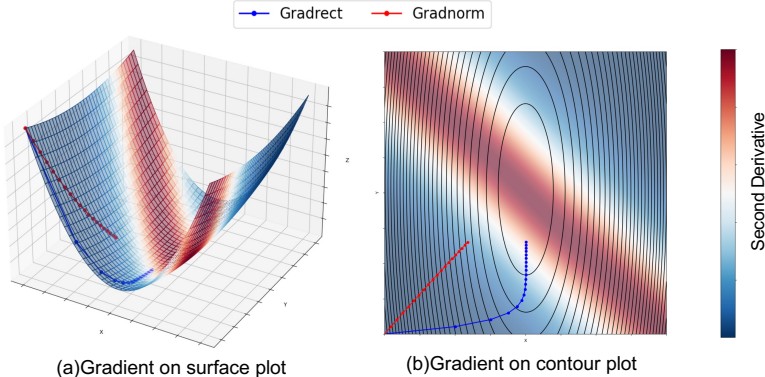

(a)Gradient on surface plot       (b)Gradient on contour plot

Figure 4: The surface and the contour plots of a toy example of loss function, which shows the difference between the gradient and the gradient after rectification of optimization. After rectified by FIM, the gradient is deviating from the directions with large fisher information values, which can be reflected in the optimization progress on the loss surface. With a fixed learning rate and update schemes, the optimization of rectified gradient can achieve faster convergence Martens (2020).

when processed by a pre-trained model. The gradient-based approaches have significant untapped potential that remains largely unexplored.

In terms of the fine-tuning methods, they typically leverage auxiliary outlier datasets for a retraining process and apply specific regularization techniques to the model therein. For example, models are encouraged to give less confidence predictions Hendrycks & Gimpel (2016); Lee et al. (2018) or higher energies for auxiliary outlier data. Recent advances in this branch mainly include improving the efficiency of leveraging auxiliary outlier data through outlier mining Chen et al. (2021); Ming et al. (2022) and virtual outlier samples synthesizing Du et al. (2022); Lee et al. (2017). Note that, the scope of this paper mainly focuses on post-hoc methods, superior over fine-tuning methods of being easy to use and general applicability without modifying the training objective. The latter property is especially desirable for the adoption of OOD detection methods in real-world production environments, when the overhead cost of retraining can be prohibitive.

**Influence Function.** Influence function is a classic method from the robust statistics literature which aims at estimating the effect of removing an individual training point on a model's parameters and corresponding predictions without the cost of retraining the model Hampel (1974); Cook (1977). The main concept of influence function is to study the impact on models through the lens of their training data. Prior work mainly study influence function on linear models and recently it gains attention in machine learning and can be efficiently approximated based on second-order optimization techniques Agarwal et al. (2016); Koh & Liang (2017). There is a wide range of application for influence function, such as explaining predictions Koh & Liang (2017), investigating model bias Brunet et al. (2019); Wang et al. (2019), detecting adversarial attacks Cohen et al. (2020) and OOD generalization problem Ye et al. (2021).

## 6    CONCLUSION

OOD detection is a important but challenging problem. We have proposed a relatively simple post-hoc OOD detection method based on rectified gradient. Specifically, we correct gradient to the directions which are more informative based on fisher information to discern the distribution difference between ID and OOD data most. We conduct a variety of ablation tests and verify our effectiveness. Additionally, we analyze the experimental results thus gain insights into the underlying reasons behind the results.

Our method gives new state-of-the-art detection results on large-scale out-of-distribution settings without requiring access to anything other than the training data itself. In the future, we will explore the usage scenarios of FIM for other scoring strategies and fine-tuning approaches in OOD detection.

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
