# OpenReview forum: "Rectifying Gradient-based OOD Detection via Fisher Information Matrix"
_ICLR.cc/2025/Conference — ICLR 2025 Conference Withdrawn Submission_

### Official Review · Reviewer_iBSJ · 2024-10-31

**Soundness:** 3
**Presentation:** 3
**Contribution:** 3
**Rating:** 6
**Confidence:** 2

**Summary:**

This paper proposes a straightforward OOD detection method based on rectified gradients, leveraging an empirical estimation of the Fisher information matrix. Comprehensive experiments are conducted to demonstrate the method's effectiveness.

**Strengths:**

1. The proposed method is easy to interpret and simple to implement.

2. The experiments are extensive, and the method’s effectiveness appears convincing.

3. The paper is well-written, with a clear and logical structure.

**Weaknesses:**

I am not an expert in the OOD detection field and would appreciate comments from other reviewers.

**Questions:**

1. In line 155, the authors state that the gradient magnitudes of ID data should be larger than those for OOD data. Isn’t it the opposite? Wouldn’t lower gradient magnitudes indicate higher confidence of the model in the sample?

2. In practice, how should one select samples to estimate the FIM? Is it better to randomly select samples with confidence scores above a certain threshold, or to use the top samples with the highest confidence scores? The latter approach requires more computation. Additionally, what would be a reasonable number of samples for this estimation?

---

### Official Review · Reviewer_3Z2q · 2024-11-03

**Soundness:** 1
**Presentation:** 1
**Contribution:** 2
**Rating:** 3
**Confidence:** 5

**Summary:**

This work builds upon existing gradient-based OOD detection methods [1, 2] by introducing the Fisher information matrix to rectify gradients. Limited experimental results demonstrate the effectiveness of their method in OOD detection tasks. Additionally, the authors show that incorporating the widely used outlier-exposure (OE) [3] can further enhance the performance of their proposed approach.

**Reference**

[1] Huang, Rui, Andrew Geng, and Yixuan Li. "On the importance of gradients for detecting distributional shifts in the wild." Advances in Neural Information Processing Systems 34 (2021): 677-689.

[2] Igoe, Conor, et al. "How useful are gradients for ood detection really?." arXiv preprint arXiv:2205.10439 (2022).

[3] Hendrycks, Dan, Mantas Mazeika, and Thomas Dietterich. "Deep anomaly detection with outlier exposure." arXiv preprint arXiv:1812.04606 (2018).

**Strengths:**

It is a simple idea that works. Empirical result supports its effectiveness.

**Weaknesses:**

I acknowledge this is a simple and good idea. However, there are still some steps to transfer these empirical results into an academic paper.

Generally speaking:

- As the method is not principled, careful justification in writing is important.
- As a purely empirical work, more experiments are needed to support solid results.

## Weaknesses of the method

### Non-theoretical approach

As mentioned, the method is not principled. The basis of this work, [1], relies on intuition and observation, using the p-norm of the gradient of the KL divergence between a discrete uniform distribution and the classifier's categorical distribution output with respect to $\theta$ as the criterion.

A non-theoretical approach is not inherently a weakness; however, such approaches typically require more thorough justification.

### Using the variance of Stein score to rectify the gradient of KL divergence

The Fisher information matrix, which is defined by the variance of the Stein score (the gradient of the log likelihood with respect to $\theta$) [2], reflects the uncertainty of the Stein score w.r.t. $\theta$. The authors want to use this uncertainty to rectify the gradient of the KL divergence, which is good, but the motivation have to be justified carefully, because a more natural way could be "using the variance of the gradient of KL divergence to rectify the gradient of KL divergence" or "using the variance of the Stein score to rectify the Stein score".Either theoretical justification or empirical justification is needed here.

It is good that the authors mentioned [3] in Eq.9, but that is all about the Stein score.

### Why is Outlier Exposure's widespread benefit surprising?

It is not surprising that Outlier Exposure [4] improves this method's performance, as it enhances the performance of many other methods as well.

## Weaknesses of justifications

It seems like the authors are trying to find some theory-like stories to support the usage of Fisher information matrix, however this does not really work.

###  The link between influence function and GradRect is not mathematical
In line 257 of the submission, the authors wrote

> Note that equation 7 exactly maximizes the influence function defined by equation 8.

I do not think so, at least they could be equivalent under some certain conditions. Please justify this and give it a proof.

From line 258 to line 262 the authors are trying to explain the Outlier Exposure method based on its loss function, it is correct but not related to the authors' contribution. As I mentioned, Outlier Exposure is good and widely used, but what is the surprise in this work?

### The Example in Figure 4 is not related to this work
I acknowledge that Natural Gradient is a good method, but what is the relationship between the Natural Gradient and this method except ... they both using the Fisher information matrix?

### Justify the contributions
I suggest moving the paragraphs about Eq. 3 and Eq. 5 to the background section since they are Fisher's work.

**Reference**

[1] Huang, Rui, Andrew Geng, and Yixuan Li. "On the importance of gradients for detecting distributional shifts in the wild." Advances in Neural Information Processing Systems 34 (2021): 677-689.

[2] https://en.wikipedia.org/wiki/Fisher_information

[3] Robert F Ling. Residuals and influence in regression, 1984.

[4] Hendrycks, Dan, Mantas Mazeika, and Thomas Dietterich. "Deep anomaly detection with outlier exposure." arXiv preprint arXiv:1812.04606 (2018).

**Questions:**

See the Weaknesses section.

**Details Of Ethics Concerns:**

N / A

---

> ### Comment · Reviewer_3Z2q · 2024-11-26
>
> The discussion deadline is near, I will keep the score if there is no response to this review.

---

### Official Review · Reviewer_8nXC · 2024-11-05

**Soundness:** 2
**Presentation:** 2
**Contribution:** 2
**Rating:** 3
**Confidence:** 2

**Summary:**

Recent works for OOD detection have an interesting focus on leveraging gradient information for the relevant tasks. This paper argues that some gradient components are lacking important information which results in performance degradation. It leads to the proposed GradRect methods, that is compared with a few works for its effectiveness.

**Strengths:**

Overall, the work is well written. The presentation is fair but not clear regarding the recent works besides the GradNorm. Some experiments have been presented for the performance gain.

**Weaknesses:**

Some concerns for the discussion and comparison for GradRect, which is the *gradient rectification* methods, are discussed. While the work aims to rectify or clip the original gradient information for more compact and informative process for OOD detection task, it remain unclear for some reasons:

1. It is unclear about the removal of uninformative gradients process. Since this is particular the key contribution in this work, I failed to find its significance to the contributions of performance gain, especially when it is aligned with the **directions** for OOD detection in gradient space. It is expected more mathematic derivations are included.

2. The training and fine-tuning for further improvement are not clear. Given an OOD distribution available for training, how does the algorithm hold its significance for OOD detection task?

3. The experimental design is outdated. Some more recent works published after 2021 are not well discussed and compared, such as 'Out-of-distribution detection with deep nearest neighbors', 'React: Out-of-distribution detection with rectified activations', 'Dream the impossible: Outlier imagination with diffusion models', 'Learning to augment distributions for out-of-distribution detection', 'Out-of-distribution detection learning with unreliable out-of-distribution sources', 'Diversified outlier exposure for out-of-distribution detection via informative extrapolation' and so on.

**Questions:**

Please refer to my discussion of weakness for consideration.

---

### Official Review · Reviewer_pHPT · 2024-11-06

**Soundness:** 2
**Presentation:** 2
**Contribution:** 2
**Rating:** 3
**Confidence:** 4

**Summary:**

The paper presents a method for OOD detection, arguing that previous gradient-based OOD detection methods are limited by over-parameterization, which leads to suboptimal performance. To address this, they propose a method called GradRect, which uses the Fisher Information Matrix to correct gradients in directions that are uninformative for detecting distribution changes. The authors connect GradRect with classical theories of identifying influential observations and suggest that model fine-tuning with outlier exposure can further enhance GradRect's performance.

**Strengths:**

1. The paper introduces GradRect, which utilizes the Fisher Information Matrix to rectify gradients for improved OOD detection, representing a novel approach in the field.
2. The ablation studies show that GradRect is robust and effective across different model architectures, which speaks to its generalizability.
3. The writting of method is clear.

**Weaknesses:**

1. Motivation is not powerful enough. The motivation presented in the paper for the proposed method appears to be somewhat underdeveloped. Specifically, Figure 2b demonstrates an intriguing inverse trend for the OOD dataset Places, contrasting with iSUN, where performance initially decreases and subsequently increases with the percentage of clipped gradients. The authors should elaborate on this observation to strengthen the rationale behind their approach. A more compelling argument would benefit the overall impact of the paper.
2. The insight is meaningless. The paper suggests that the presence of uninformative components in gradients is a novel insight. However, this concept is not entirely new. For instance, Reference [R1] introduces an orthogonal projection onto gradient subspaces, and Reference [R2] explores the attribution of gradients, both of which have been shown to enhance OOD detection performance through gradient rectification techniques. The authors should acknowledge these related works and discuss how their approach differs and contributes uniquely to the field.
3. Comparison is outdated. The paper's comparison with existing methods appears to be somewhat outdated. Given the rapid advancements in the field, it is crucial for the authors to include and compare their method with the latest techniques published in 2024. This will ensure that the contributions of the paper are assessed within the current state-of-the-art and highlight the innovative aspects of their work.
4. Limited discussion on over-parameterization. The paper would benefit from a more in-depth exploration of the role of over-parameterization in OOD detection and how the proposed GradRect method specifically addresses these challenges. A thorough discussion on this topic will provide a better understanding of the underlying issues and the effectiveness of GradRect in mitigating them.

[R1] Behpour, Sima, et al. "GradOrth: a simple yet efficient out-of-distribution detection with orthogonal projection of gradients." Advances in Neural Information Processing Systems 36 (2024).

[R2] Chen, Jinggang, et al. "GAIA: delving into gradient-based attribution abnormality for out-of-distribution detection." Advances in Neural Information Processing Systems 36 (2023): 79946-79958.

**Questions:**

1. The paper claims that using the Fisher Information Matrix to correct uninformative gradients in OOD detection offers certain benefits. However, the theoretical underpinnings of this claim are not sufficiently developed. The authors should provide a rigorous theoretical proof to substantiate their method's effectiveness and explain why it is superior to existing approaches
2. The authors assert that GradRect exhibits greater stability, as shown in Figures 1c and 1d. However, these figures primarily present performance comparisons rather than a comprehensive analysis of stability. A more robust demonstration of stability would involve comparing GradRect with other methods across various aspects, such as different percentages of clipped gradients as depicted in Figures 1a and 1b. This additional analysis would strengthen the paper's claims regarding the stability of their method.
3. While the authors mention that the Fisher Information Matrix and its inverse can be precomputed, the paper lacks detailed information on the computational overhead compared to other methods, particularly for large-scale applications. High latency could limit the practical applicability of the method in real-world scenarios. The authors should provide a comprehensive analysis of the computational costs associated with their method and discuss any potential strategies to mitigate these costs.

---

### Official Review · Reviewer_m7NM · 2024-11-07

**Soundness:** 3
**Presentation:** 3
**Contribution:** 3
**Rating:** 6
**Confidence:** 4

**Summary:**

This work proposes a new method, GradRect, for OOD detection. The idea is to use the inverse Fisher Information Matrix to remove unimportant information encoded in gradients (specifically GradNorm) so as to improve detection rates. Experiments on (relatively old) CIFAR and ImageNet benchmarks demonstrate improved detection performance. Sufficient ablations are conducted to validate the effectiveness and/or robustness of the proposed method.

**Strengths:**

1. The proposed method makes sense: Given abundant information encoded in gradients, removing irrelevant information and focusing on important cues is necessary.
2. The paper did a good job connecting GradRect with theories and interpreting it from different perspectives (Eqn 4's motivation, Influence Function, relationship with gradient descent).

**Weaknesses:**

My major concern is with the used benchmarks. Specifically, there is no near-OOD datasets considered (e.g., CIFAR-10 v.s. CIFAR-100, ImageNet v.s. NINCO/SSB; see OpenOOD [1] for details), while near-OOD detection has been recognized as a more challenging and meaningful task in the field [1,2]. In addition, the used LSUN-Resize benchmark for CIFAR-10 might be problematic (exhibiting resizing artifacts), as pointed out by [3].

I suggest adding at least one near-OOD dataset in each setting.

[1] OpenOOD v1.5: Enhanced Benchmark for Out-of-Distribution Detection
[2] Detecting Semantic Anomalies
[3] CSI: Novelty Detection via Contrastive Learning on Distributionally Shifted Instances

**Questions:**

1. It is unclear what exactly is the "proportion of rectified gradient" in Sec. 4.3. I guess one adjusts such proportion by masking elements in the inverse of FIM? If so, how to choose which elements to mask when doing the experiment?

---

### Note · Authors · 2024-11-27

**Comment:**

We are grateful for the time and thorough feedback provided by the reviewers, which will be valuable to enhance our work. However, we have decided to withdraw this paper from consideration for the conference. Once again, we thank you for your thoughtful and constructive comments.

**Withdrawal Confirmation:**

I have read and agree with the venue's withdrawal policy on behalf of myself and my co-authors.